

# Local and relaxed clocks: the best of both worlds

Mathieu Fourment and Aaron E. Darling

ithree institute, University of Technology Sydney, Sydney, Australia

## ABSTRACT

Time-resolved phylogenetic methods use information about the time of sample collection to estimate the rate of evolution. Originally, the models used to estimate evolutionary rates were quite simple, assuming that all lineages evolve at the same rate, an assumption commonly known as the molecular clock. Richer and more complex models have since been introduced to capture the phenomenon of substitution rate variation among lineages. Two well known model extensions are the local clock, wherein all lineages in a clade share a common substitution rate, and the uncorrelated relaxed clock, wherein the substitution rate on each lineage is independent from other lineages while being constrained to fit some parametric distribution. We introduce a further model extension, called the flexible local clock (FLC), which provides a flexible framework to combine relaxed clock models with local clock models. We evaluate the flexible local clock on simulated and real datasets and show that it provides substantially improved fit to an influenza dataset. An implementation of the model is available for download from https://www.github.com/4ment/flc.

## INTRODUCTION

Phylogenetic methods provide a powerful framework for reconstructing the evolutionary history of viruses, bacteria, and other organisms. Correctly estimating the rate at which mutations accumulate in a lineage is essential for phylogenetic analysis, as the accuracy of inferred rates can heavily impact other aspects of the analysis. Classic approaches to infer the substitution rate of a group of organisms rely on the existence of a so-called "molecular clock". The molecular clock hypothesis dictates that mutations accumulate at an approximately steady rate over time, implying that the genetic distance between two organisms is proportional to the time since these organisms last shared a common ancestor. The molecular clock hypothesis was first proposed almost 50 years ago by Emile Zuckerkandl and Linus Pauling (*Zuckerkandl & Pauling, 1965*) who suggested that the substitution rate was effectively constant over time. This very restricted model of evolution has been implemented using a "strict clock" model in phylogenetic inference software, but the rates of evolution in many organisms appear to change over time and the model can not capture this phenomenon. Many prior studies have shown evidence of rate variation among species (*Wu & Li, 1985*; *Woolfit & Bromham, 2003*; *Fourment & Holmes, 2015*), especially among highly divergent taxa. Rate variation can be attributed to a range of

Corresponding author
Mathieu Fourment,
mathieu.fourment@uts.edu.au

factors including difference in background mutation rate, generation time, population size, and natural selection (*Bromham, 2009*; *Duchêne et al., 2016*). The accurate inference of the substitution rate along a phylogeny has played an important role in estimating the timing of emergence and the geographic provenance of viral and bacterial outbreaks (*Vijaykrishna et al., 2008*; *Holmes et al., 2016*). This led to the development of more realistic molecular clock models. These more realistic models come with the expense of added complexity, but an increase in computational power in recent years has made inference feasible under complex models.

## Relaxed clock models

Richer models can better capture the complexity of the evolutionary process. *Thorne, Kishino & Painter (1998)* and *Sanderson (2002)* proposed to model rate heterogeneity among lineages using auto-correlated clock models using penalized likelihood and Bayesian inference, respectively. In these parameter rich models, the substitution rate of each lineage is assumed to be correlated with that of the parent lineage, emphasizing a gradual rate change between neighboring lineages. The auto-correlation assumption could be justified by considering that the substitution rate is influenced by heritable mechanisms such as metabolic rate or generation time (*Gillespie, 1994*). In this framework, it is assumed that the substitution rate varies gradually between neighboring lineages. However there is no guarantee that rates evolve in an auto-correlated manner, especially when the timescale under study is relatively small (*Drummond et al., 2006*). An alternative approach is to assume that substitution rates on adjacent branches are independent draws from an underlying parametric distribution. *Drummond et al. (2006)* chose to forgo the hierarchical Bayesian framework and opted for a likelihood approach that requires the rates to fit a discretized distribution, implemented in the BEAST software. The log-normal and exponential distributions are commonly used to model the rate heterogeneity among lineages, as they are available in the widely used BEAST1 and BEAST2 packages (*Drummond & Rambaut, 2007*; *Bouckaert et al., 2014*). Alternatively, *Lepage et al. (2007)* proposed to model the rate process as a pure white noise process with a gamma distribution as its stationary distribution. The auto-correlated and uncorrelated clock models are referred to as *relaxed clock models* due their ability to relax the constant rate assumption.

## Local clock models

Local clock models are an alternative to relaxed clocks, where the model assumes that the substitution rate is constant within a clade but can differ between clades (*Yoder & Yang, 2000*; *Yang & Yoder, 2003*). Local clocks are based on the assumption that the molecular clock hypothesis holds for closely related species. For example, this type of clock can be useful for gene trees that have dense taxon sampling for which rate variation among related lineages is expected to be minimal (*Drummond et al., 2006*). However, assigning local clocks on a phylogeny is not without difficulty in the absence of a preliminary analysis, and they warrant compute intensive methods that usually require the topology to be fixed (*Yang, 2004*; *Aris-Brosou, 2007*; *Fourment & Holmes, 2014*). *Drummond & Suchard (2010)* developed a Bayesian algorithm that infers the phylogeny along with the

number and location of the local clocks directly from the data. Many other methods have been developed to model rate variation among lineages such as the compound Poisson process (*Huelsenbeck et al., 2001*), the Dirichlet process prior (*Heath, 2012*) and the autocorrelated CIR model (*Lepage et al., 2007*). For a broader overview on the topic we direct the reader to reviews (*Ho & Duchêne, 2014*; *Dos Reis, Donoghue & Yang, 2016*) and performance benchmark studies (*Ho et al., 2005*; *Lepage et al., 2007*).

### Introducing the flexible local clock

In this manuscript we introduce a hybrid model that integrates features of both the local and the relaxed clock models. In the model each local clock can be specified either as a strict clock (as in the original formulation of the local clock model) or as a relaxed clock. Specifically, this approach allows closely related lineages to be modeled with a single substitution rate (i.e., strict clock) while other lineages in the same phylogeny with significant rate variation can be described with a more flexible model (i.e., relaxed clock). We call this model the flexible local clock (FLC) model. The FLC model is similar to previous models (*Yang, 2004*; *Aris-Brosou, 2007*; *Fourment & Holmes, 2014*) as it requires to *a priori* define the number and location of the local clocks but it allows a richer description of the rate process trajectory. We evaluate the FLC model using a newly implemented module for the BEAST2 package, which uses Markov chain Monte Carlo to carry out inference of model parameters (*Bouckaert et al., 2014*). We reanalyzed an influenza virus (*Drummond & Suchard, 2010*) and a HIV (*Wertheim, Fourment & Kosakovsky Pond, 2012*) data set to evaluate the utility of the FLC model and compared its fit to the data to that given by other models.

## METHODS

Phylogenetic packages such as BEAST provide several options to model lineage-specific rate variation, known as heterotachy, without overfitting the model. One of the first ingredients of the FLC model is the uncorrelated relaxed clock model (*Drummond et al., 2006*), arguably the most popular lineage-specific rate model. The uncorrelated relaxed clock model uses a single discretized parametric distribution to model rate heterogeneity. In the original formulation of the model, a parametric distribution, usually lognormal, is discretized into a fixed number of components, with the number of these components equal to the number of branches $b$ in the tree. In its simplest form, the model assumes a one-to-one relationship between a rate at a branch and one of the components. For a lognormal distribution, this approach only requires estimating two parameters (i.e., mean and standard deviation) instead of $2N - 2$ parameters if a hierarchical model was used, where $N$ is the number of sequences. As in the standard relaxed clock model (*Drummond et al., 2006*), the FLC model can use the exponential distribution to model rate variation, although our model could also use other appropriate parametric distributions.

The other ingredient of the FLC model is the local clock which was first proposed by *Yoder & Yang (2000)*. This model allows lineages within a region of the tree to evolve at exactly the same rate. We define a local clock on a phylogeny as a monophyletic group where the substitution rate of every lineage is equal. As in *Drummond & Suchard (2010)*,

we assume the existence of another clock (e.g., a 'global' clock) for lineages that are not assigned a local clock.

Herein, we propose to relax the constraint that lineages within a local clock evolve at exactly the same rate by replacing this implicit strict clock by a relaxed clock.

We applied the FLC model to two data sets of heterochronous viral nucleotide sequences. The first data set comprises an alignment of 69 human influenza A/H3N2 virus haemagglutinin (HA) sequences (987 nt in length) isolated between 1981 and 1998. The evolutionary rates and time to the most recent ancestors (tMRCAs) of this data set was previously investigated using a random local clock method (*Drummond & Suchard, 2010*) with a Bayesian Markov chain Monte Carlo (MCMC) approach implemented in BEAST1 (*Drummond & Rambaut, 2007*). We reanalysed the data using BEAST2 with each of the FLC, uncorrelated lognormal relaxed clock (UCLN), local clock (LC), and random local clock (RLC) models. As in the original study, our analyses use the HKY$+\Gamma_4$ substitution model that incorporates gamma-distributed rate variation among sites (four rate classes). The FLC and LC models require manual assignment of each lineage to a local clock with the appropriate constraints on the phylogeny. *Drummond & Rambaut (2007)* noticed that the substitution rate of the lineages comprising viruses sampled after 1990 appeared higher than the pre-1990 lineages. We therefore assigned sequences sampled after 1990 to a local clock for both LC and FLC models. For each local-based model, we conducted two separate analyses in which the branch subtending the clade containing the late viruses (1990-onward) were assigned either to a local clock or the ancestral rate. We specified a diffuse prior on the substitution rates of the LC and FLC models using an exponential distribution with a mean of 0.003. For the log-normal distribution of the relaxed clock we used an exponential prior distribution ($\lambda = 1/0.003$) on the mean parameter and an exponential prior distribution ($\lambda = 1/0.33$) on the standard deviation parameter. As in the study describing the RLC model we used a Poisson distribution with $\lambda = \log 2$ as a prior on the number of local clocks, thereby placing 50% prior probability on a single rate across the phylogeny. Finally, we assumed *a priori* that rate multipliers are independently gamma distributed with $\alpha = 0.5$ and $\beta = 2$ as in *Drummond & Suchard (2010)*.

For each data set, we calculated the marginal likelihood of the data under each model using the stepping stone algorithm to compare competing models (*Xie et al., 2011*). We used a series of 100 power posteriors where $\beta$ values are chosen to be evenly spaced quantiles of a Beta distribution with parameters $\alpha = 0.3$ and $\beta = 1.0$. These parameters result in half of the power posteriors being evaluated for $\beta < 0.1$ for which the power posterior is changing rapidly, as suggested by *Xie et al. (2011)*. Each MCMC was run for 10 million iterations and the first 10% of the samples were discarded as burn-in.

## Simulations

To validate the implementation of our model we simulated data sets using the FLC model. Our approach is similar to a simulation-based study (*Worobey, Han & Rambaut, 2014*) that showed that the local clock model is best suited to model rate variation among influenza virus sequences sampled from three different hosts (i.e., equine, human, and

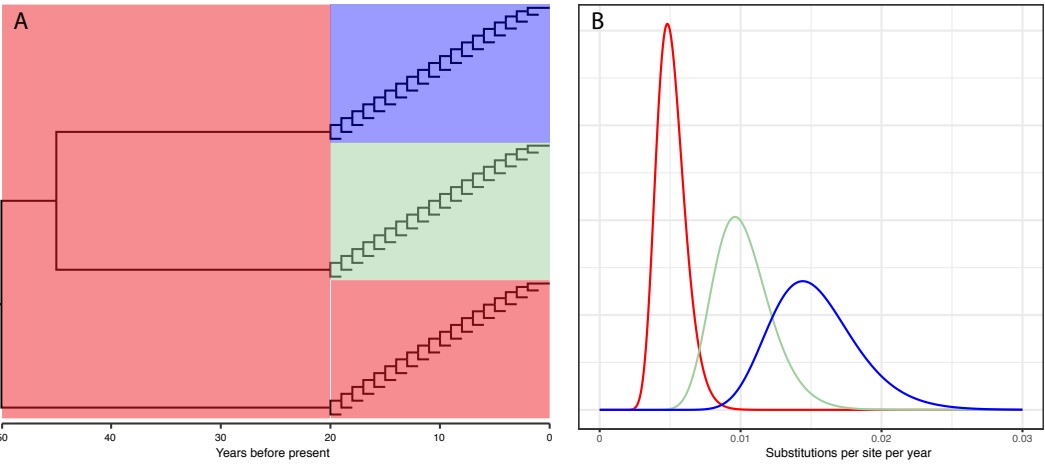

**Figure 1** (A) Phylogenetic tree and (B) substitution rates used to simulate data sets.

birds). Worobey et al. assigned different local clocks to each of the monophyletic bird and human clades and simulated nucleotide alignments containing 10,000 sites. Phylogenies were estimated using either a strict, flexible local or local clock model using the BEAST2 package (*Drummond & Rambaut, 2007*). The simulations showed that only the local clock model was able to recover the true tree. In this study, we used the same topology and divergence times, and replaced standard local clocks with flexible local clocks.

Ten replicates containing 10,000 sites were simulated using the program simultron (*Fourment & Holmes, 2014*) under the HKY model ($\kappa = 3$ and equal nucleotide frequencies). The standard deviation $\sigma$ of the lognormal distributions were all set to be equal to 0.2. The $\mu$ parameter of the lognormal distributions of the equine, human, and bird clades were set such as the mean of the distributions were $5 \times 10^{-3}$, $1 \times 10^{-2}$, and $1.5 \times 10^{-2}$, respectively (Fig. 1). The choice of the parameters results in roughly bell shaped distributions centered on the substitution rates used in the Worobey et al. study. We analyzed the simulated data sets with the HKY model and the skyline coalescent tree prior under the strict, flexible local, local, relaxed, and random local clocks. The simulation script is available from http://www.github.com/4ment/flc-data.

## RESULTS AND DISCUSSION

We analyzed the influenza virus data set with BEAST2 under a variety of models including the FLC model. Since the flexible local clock can be composed of a combination of strict and relaxed clocks, we specify the type of clock between brackets. For example, we use FLC [strict&UCLN] to denote a flexible local clock with a strict clock on the early lineages (i.e., sequences before 1990) and an uncorrelated lognormal relaxed clock (UCLN) on the later lineages. For local and flexible local clocks we can specify whether the branch leading to the clade with a local clock should be included in the new clock (contains the stem). To test which models better fit to the data we calculated the marginal likelihood for each model (Table 1).

**Table 1  Marginal likelihoods calculated using the stepping stone algorithm.** UCLN, uncorrelated log-normal relaxed clock; RLC, random local clock; FLC, flexible local clock; LC, local clock. The "Contains stem" column specifies whether the branch subtending the post-1990 clade is assigned to the local clock.

| Model | Marginal likelihood | Contains stem |
|---|---|---|
| FLC [strict&UCLN] | −4381.72 | No |
| FLC [UCLN&UCLN] | −4382.07 | Yes |
| FLC [strict&UCLN] | −4382.92 | Yes |
| FLC [UCLN&UCLN] | −4383.28 | No |
| UCLN | −4385.04 | NA |
| LC | −4386.81 | Yes |
| LC | −4387.69 | No |
| RLC | −4415.36 | NA |

As in the original study (*Drummond & Suchard, 2010*), every model shows a substitution rate increase in sequences sampled after the 1990 (Fig. 2).

The marginal likelihood estimates (Table 1) suggest that the best models are the FLC models, followed by the UCLN, LC, and RLC models. The inclusion of the stem in the FLC and LC models appears to have a minor effect on the model fit depending on the model, but the marginal likelihood estimates are subject to Monte Carlo error and caution should be exercised in order to avoid overinterpreting small differences. The 95% highest posterior density (HPD) of the standard deviation of the lognormal distribution assigned to the global clock includes zero, suggesting that there is little rate variation outside the post-1990 clade (i.e., FLC [UCLN&UCLN]). It is therefore no surprise that the marginal likelihoods of the FLC models with a strict or UCLN clock on the pre-1990 lineages are similar. Interestingly, the UCLN model appears to fit better to the data than the RLC and LC models.

## Results on simulated data

We simulated 10 data sets under the flexible local model and estimated the phylogenies using several clock models. The comparison of the maximum clade credibility (MCC) tree to the true topology reveals that the strict and relaxed clock models could not recover the rooting of the true tree in any replicate. Interestingly the 95% HPD intervals of the root node age contained the true value in four and two of the replicates using the relaxed and strict clock models, respectively. The MCC trees of the standard local clock model recovered the true rooting and the root age was recovered in the 95% HPD in only three replicates. The MCC trees of the flexible local clock model had the same rooting as the true tree in 9 out of 10 cases and the 95% HPD of the root age contained the true value for 8 out of 10 replicates.

## Limitations and conclusions

As in the standard local model, the flexible local clock model introduced in this paper assumes that the user knows the number and the location of the rate shifts in the phylogeny. *Drummond & Suchard (2010)* devised the random local clock to address this limitation using a stochastic search variable selection method to sample over random local clocks.

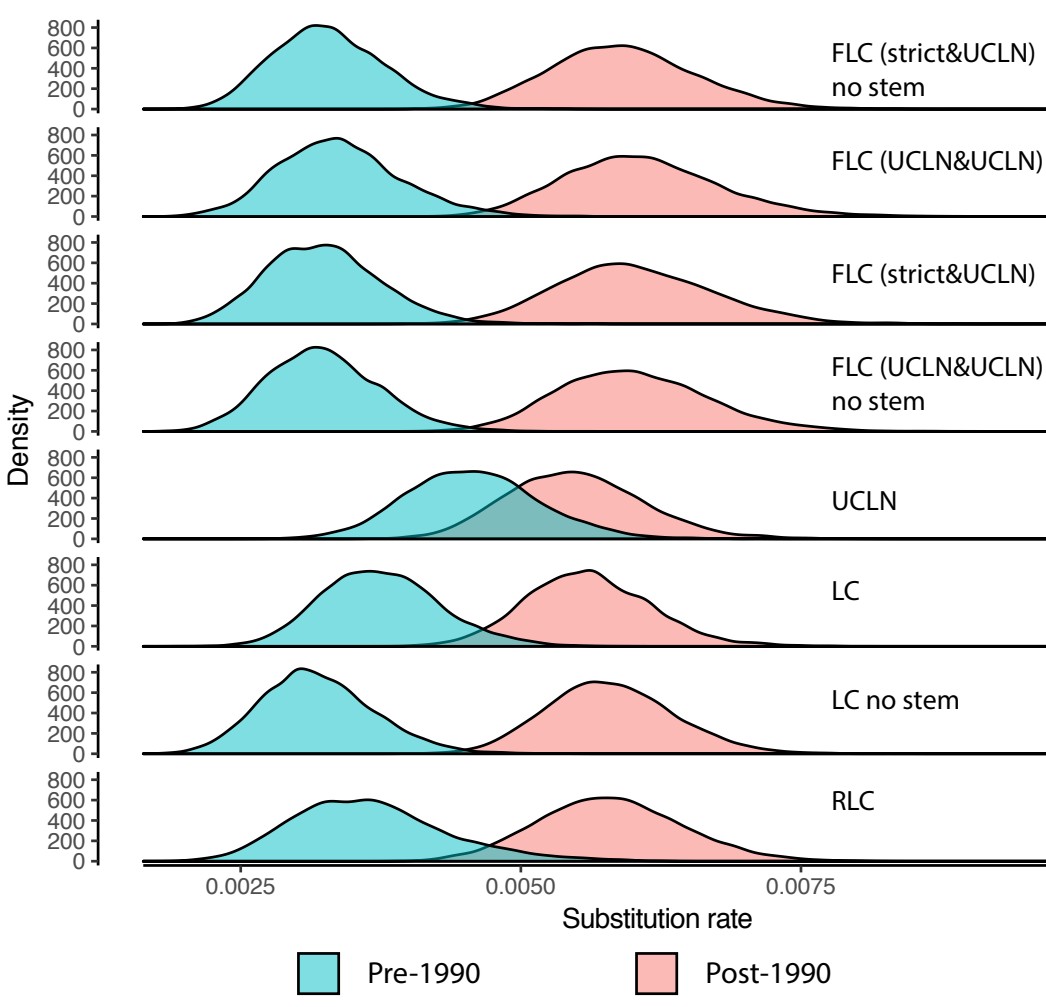

**Figure 2** **Posterior distributions of the mean substitution rate of the lineages comprising viruses sampled after 1990 and before 1990.** UCLN, uncorrelated lognormal relaxed clock; RLC, random local clock; FLC, flexible local clock; LC, local clock. For the local clock models labeled no stem, the branch subtending the post-1990 clade is not assigned to the local clock.

Unfortunately that approach is not easily amenable to integration with the FLC model since the substitution rate within a clock can either be constant or heterogeneous across lineages. Although it should be possible to use reversible jump MCMC to sample the posterior distribution it is not clear how to deal with a variable number of lineages assigned to a relaxed clock. For example, the assignment of a relaxed clock with a two parameter distribution to a single branch would over-parametrize the model. An interesting direction for further research would be to develop an algorithm that automatically selects the clock type for each local clock.

Until a method exists for automatic determination of the placement and type of local clocks, users of the model will need to determine these manually via exploratory analysis of datasets. Future work could investigate the conditions under which simpler clock models

break down and provide guidance to users of the FLC on how to identify conditions where the model is likely to provide an improved fit relative to other alternatives.

The FLC model is implemented in the BEAST2 package as a plugin and is available from https://www.github.com/4ment/flc. This implementation inherits the flexibility of the BEAST2 architecture as it can be integrated through the plugin system without changing the code base and it is fully compatible with the other model components.

### Funding
This work was supported by the Australian centre for genomic epidemiological microbiology (AusGEM). The funders had no role in study design, data collection and analysis, decision to publish, or preparation of the manuscript.

### Grant Disclosures
The following grant information was disclosed by the authors:
Australian centre for genomic epidemiological microbiology (AusGEM).

### Competing Interests
The authors declare there are no competing interests.

### Author Contributions
- Mathieu Fourment conceived and designed the experiments, performed the experiments, analyzed the data, contributed reagents/materials/analysis tools, prepared figures and/or tables, authored or reviewed drafts of the paper, approved the final draft.
- Aaron E. Darling contributed reagents/materials/analysis tools, approved the final draft.

### Data Availability
GitHub: https://www.github.com/4ment/flc.

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
