# Peer review of "Local and relaxed clocks: the best of both worlds"

_PeerJ, doi:10.7717/peerj.5140_

## Round 0.1 · original submission · Minor Revisions

Two of the reviews have recommend the paper be accepted, one has suggested that major revisions are required. My feeling is that the paper could be improved by a revision, but feel it is more in the category of minor revision, so please look at the comments of reviewer 2 and engage with their suggestions.

In particular helping those less familiar with the subject, see “The description of molecular clock hypothesis should be extended. As it stands, it is only accessible for readers familiar with the field. The concept of auto-correlation is particularly briefly explained. Similarly, the order of appearance of each of the hypotheses is rather arbitrary. Since local clocks contain less rates than uncorrelated clocks, it seems like they should appear first. It also seems appropriate to acknowledge the pure white-noise model of Lepage et al. (2007), which is the only truly uncorrelated clock model.”.

Sorry about the length of time the paper spent in review, it was not that easy to find reviewers.

Reviewer 1 ·

Basic reporting

1] Basic Reporting – The manuscript is well-written: concise, clear and professional. It deals with a specialized area, but a wider audience will be able to understand the main points of the paper.
The subject is clearly introduced. To be redundant: this is a narrow topic, but the authors have done an adequate job of describing the general problem and their approach to improving the available analytical tools.
The figures and table are clear and can be understood. There is no problem with the raw data or with the availability of the authors’ implementation of their software within BEAST.

Experimental design

2] Experimental Design – I find that the subject is within the scope of the journal. However, and this might be repetitious, the problem addressed is narrow and the authors are improving the analytical software. The biggest deficit of the manuscript, in my opinion, is that there is NO biology! Yes, influenza sequences are run through their flexible local clock software. The authors find that the resulting “fit” is slightly better (higher marginal maximum likelihood values), but it is impossible for the reader to discern if this better “number” means something in terms of influenza virus sequence evolution. This is a common situation, and the authors should not be held to a different standard than the hundreds of other articles in this area.
Again, I must stress that the manuscript is clear and that the research merits publication in PeerJ. The methods and results are clearly presented and there is sufficient detail with specific regard to the specific issue under investigation.

Validity of the findings

3] Validity of the Findings – The data/results are sound, although the improvement in marginal maximum likelihood values are not earth-shattering. The conclusion that flexible local clocks would yield an improved fit to the sequence analysis is warranted. The manuscript is well-written in terms of focus and ensuring that everything ties back to the specific question being tested.

Additional comments

4] Overall Assessment – This is a well-written manuscript that is focused and tightly structured. Publication is recommended. At the same time, the authors will in the future – one hopes – integrate their modeling results into the broader question of sequence evolution.

Reviewer 2 ·

Basic reporting

The pre-print of this article has been discussed with multiple colleagues, and we agree that the piece of research is mostly well presented and clear. Since this piece includes a package, the readership will largely be broader than specialists and other model-developers. For this reason, I have several comments on the basic reporting.

The description of molecular clock hypothesis should be extended. As it stands, it is only accessible for readers familiar with the field. The concept of auto-correlation is particularly briefly explained. Similarly, the order of appearance of each of the hypotheses is rather arbitrary. Since local clocks contain less rates than uncorrelated clocks, it seems like they should appear first. It also seems appropriate to acknowledge the pure white-noise model of Lepage et al. (2007), which is the only truly uncorrelated clock model.

The description of the flexible local clock should also be more detailed to reach non-specialists. The fact that the number and location of clocks must be determined before the analysis should be mentioned explicitly from the outset. The parametric distributions of rates across branches that are permitted in the author's implementation should also be outlined. Also, are all the other default model settings (substitution models, tree priors) in BEAST2 also allowed in the add on?

Other than the technical details of the proposed model, there is a clear lack of introduction of the biological interpretation of the model. In which scenarios do we expect abrupt changes in the rate, followed by also possibly abrupt changes within each local clock. Are there any examples of this in nature. Or perhaps more intuitively the authors could say why the existing clock models are unrealistic, and how these gaps are met in the new model.

The paragraph starting on page 79 should be broken down into data sets at least, because it is difficult to follow the point in the present form.

The description of the simulations study requires more detail. For example, a non-expert will be lost after reading sentence 119, which refers to non-specific lognormal distributions.

The description of the package in the GitHub repository is extremely brief. A step-by-step process of installing the package would be helpful, as well as a brief tutorial. Also note that in many machines the Library folder is hidden, making it necessary for users to install the package through the command line. Once it is installed, the template does not automatically appear in BEAST2, so the step-by-step process of installation is fundamental.

It would me more suitable to report the pairwise Bayes Factors between models than the marginal likelihood estimates. It is clear that the difference in marginal likelihoods is small, so the Bayes Factors will provide a clearer representation of the improvement in model fit.

The proposed model is substantially more parameter-rich than any of the other models discussed in the introduction. This can come at a substantial computational cost. Reporting the difference in computational demand across models would help users decide whether it is worth using this model.

Lepage, T., Bryant, D., Philippe, H., & Lartillot, N. (2007). A general comparison of relaxed molecular clock models. Molecular biology and evolution, 24(12), 2669-2680.

Experimental design

In its present form, the manuscript gives little insight into the difference between the existing local clock model and the new flexible local clock.

The new model could be tested much more thoroughly to determine exactly when it is more powerful than other clock models. Presumably it will be more powerful than the local clock when the variance in the rate is high within each local clock. Showing how much variance is needed for the standard local clock to fail would be interesting, as would be testing the impact of different magnitudes of change. Presumably the uncertainty in rate estimates is also greater when using the flexible local clock, making it less precise but more accurate. This follows the bias-variance tradeoff in statistics, and would give readers an insight into how this model fits with regards to other existing models.

The simulations at present are of little use and are only briefly discussed. It would be helpful to be explicit about why they have been included and what they say about the model. A figure with the simulation results would be greatly insightful, for example in the form of a boxplot showing the sizes of the HPDI of the mean rate or the rate estimates across scenarios.

Validity of the findings

The new method presente is a valid and natural extension of existing methods. However, it is possible to be more explicit about the impact that this new model will have when analysing empirical data.

The Limitations and Conclusions section should outline how the model might not be biologically realistic, and any possible extensions that could lead to improvement.

Additional comments

The idea for this clock model is interesting as long as it is well justified biologically. I think this model is a useful and interesting step, but it requires a more thorough justification and demonstration of its power.

The authors have made a reasonable effort in making this new model accessible and showing that it can be useful. However, both of these points should be extended. This would make the method truly accessible and appealing for non-specialists and will give a clear picture of the model's performance for specialists willing to expand or examine clock models in the future.

Reviewer 3 ·

Basic reporting

No comment.

Experimental design

No comment.

Validity of the findings

No comment.

Additional comments

The manuscript entitled “Local and relaxed clocks, the best of both worlds” describes an implementation for the widely used BEAST software that allows the use of relaxed clocks within a context of specifying local locals (that previously used only strict local clocks). In my opinion it is a smart implementation that I am certain will be used in multiple situations and different types of data. The manuscript is clearly written in terms of objectives, methodologies and testing of the model (real data and simulated data) and it is clear about its limitations. Given this I advise for its publication.

---

## Round 0.2 · accepted · Accept

The rebuttal is well thought out and the paper improved, let make this plugin available to the field for additional testing!

#